# Time-Resolved Profiling Reveals ATF3 as a Novel Mediator of Endocrine Resistance in Breast Cancer

**DOI:** 10.3390/cancers12102918

**Published:** 2020-10-11

**Authors:** Simone Borgoni, Emre Sofyalı, Maryam Soleimani, Heike Wilhelm, Karin Müller-Decker, Rainer Will, Ashish Noronha, Lukas Beumers, Pernette J. Verschure, Yosef Yarden, Luca Magnani, Antoine H.C. van Kampen, Perry D. Moerland, Stefan Wiemann

**Affiliations:** 1Division of Molecular Genome Analysis, German Cancer Research Center (DKFZ), Im Neuenheimer Feld 580, 69120 Heidelberg, Germany; esofyali@gmail.com (E.S.); H.Wilhelm@dkfz-heidelberg.de (H.W.); l.beumers@dkfzheidelberg.de (L.B.); 2Faculty of Biosciences, University of Heidelberg, Im Neuenheimer Feld 234, 69120 Heidelberg, Germany; 3Bioinformatics Laboratory, Department of Clinical Epidemiology, Biostatistics, and Bioinformatics, Amsterdam Public Health Research Institute, Amsterdam UMC, University of Amsterdam, Meibergdreef 9, 1105 AZ Amsterdam, The Netherlands; M.SoleimaniDodaran@uva.nl (M.S.); a.h.vankampen@amc.uva.nl (A.H.C.v.K.); p.d.moerland@amsterdamumc.nl (P.D.M.); 4Biosystems Data Analysis, Swammerdam Institute for Life Sciences, University of Amsterdam, Science Park 904, 1098 XH Amsterdam, The Netherlands; 5Tumor Models Core Facility, German Cancer Research Center (DKFZ), Im Neuenheimer Feld 280, 69120 Heidelberg, Germany; k.mueller-decker@dkfz-heidelberg.de; 6Genomics and Proteomics Core Facility, German Cancer Research Center (DKFZ), Im Neuenheimer Feld 280, 69120 Heidelberg, Germany; r.will@dkfz-heidelberg.de; 7Department of Biological Regulation, Weizmann Institute of Science, 7610001 Rehovot, Israel; ashish.noronha@weizmann.ac.il (A.N.); yosef.yarden@weizmann.ac.il (Y.Y.); 8Synthetic Systems Biology and Nuclear Organization, Swammerdam Institute for Life Sciences, University of Amsterdam, Science Park 904, 1098 XH Amsterdam, The Netherlands; P.J.Verschure@uva.nl; 9Department of Surgery and Cancer, Imperial College London, W12 0NN London, UK; l.magnani@imperial.ac.uk

**Keywords:** breast cancer, endocrine therapy, drug-resistance, ATF3, RPPA

## Abstract

**Simple Summary:**

Breast cancer is one of the leading causes of death for women worldwide. Patients whose tumors express estrogen receptor α (ERα) are the vast majority and are mostly treated with targeted endocrine therapy. However, resistance development poses an urgent clinical problem and the mechanisms of this phenomenon are not fully understood. In this study we identified ATF3 as a novel regulator of the response to therapy via rewiring of central signaling processes towards the adaptation to endocrine treatment. Our work adds a new piece to the complex puzzle of the molecular mechanisms at the basis of endocrine therapy resistance and proposes ATF3 and the downstream pathways as putative targets for novel combinatorial treatment strategies.

**Abstract:**

Breast cancer is one of the leading causes of death for women worldwide. Patients whose tumors express Estrogen Receptor α account for around 70% of cases and are mostly treated with targeted endocrine therapy. However, depending on the degree of severity of the disease at diagnosis, 10 to 40% of these tumors eventually relapse due to resistance development. Even though recent novel approaches as the combination with CDK4/6 inhibitors increased the overall survival of relapsing patients, this remains relatively short and there is a urgent need to find alternative targetable pathways. In this study we profiled the early phases of the resistance development process to uncover drivers of this phenomenon. Time-resolved analysis revealed that ATF3, a member of the ATF/CREB family of transcription factors, acts as a novel regulator of the response to therapy via rewiring of central signaling processes towards the adaptation to endocrine treatment. ATF3 was found to be essential in controlling crucial processes such as proliferation, cell cycle, and apoptosis during the early response to treatment through the regulation of MAPK/AKT signaling pathways. Its essential role was confirmed in vivo in a mouse model, and elevated expression of *ATF3* was verified in patient datasets, adding clinical relevance to our findings. This study proposes ATF3 as a novel mediator of endocrine resistance development in breast cancer and elucidates its role in the regulation of downstream pathways activities.

## 1. Introduction

Breast cancer is the most diagnosed type of cancer among women and the second-leading cause of death after lung cancer [1]. It is a highly heterogeneous disease, commonly classified either by intrinsic subtyping, which is based on gene expression classifiers (Luminal A (LA), Luminal B (LB), Her2-enrichded (HER2+), basal-like (Basal), and normal-like (Normal) [2,3] or by clinical sub-typing, based on immunohistochemistry analysis of estrogen receptor (ER), progesterone receptor (PR), HER2 receptor (HER2) and Ki-67 as well as FISH-analysis for HER2 amplification detection [4]. These markers allow to stratify the tumors in treatment specific groups: “LA like” (ER+/HER2−/Ki67−), “LB like” (ER+/HER2−/Ki67+ or ER+/HER2+), “HER2+ not luminal” (ER−/PR−/HER2+) and “Triple negative” (TN; ER−/PR−/HER2−) [5]. Luminal tumors are the most abundant, accounting for around 70% of all breast cancers, for which endocrine therapy is the standard treatment [6,7]. Endocrine therapy consists of different drugs that act on ER activation, including selective ER modulators (SERMs) and downregulators (SERDs), like tamoxifen and fulvestrant, respectively, and aromatase inhibitors (AIs), that block the enzymes involved in the synthesis of estrogens [8].

Despite the clear benefit of endocrine therapy for patients with ER+ breast cancer, resistance to treatment is a critical clinical issue that affects approximately 10–15% of patients in the first five years [9] and around 20–25% within 15 years from the start of the treatment [10,11,12]. The 20 years risk of recurrence has recently been described to be highly dependent on the tumor diameter and nodal status at diagnosis, ranging from as low as 10% for low-grade diseases up to 41% in advanced diseases with more than four nodes involved [13]. After the failure of first line endocrine therapy, recurrent ER+ breast cancer can be treated with alternative strategies to target growth and survival pathways. Common strategies employ fulvestrant to block and degrade the ER both in monotherapy or in combination with other approved drugs for advanced breast cancer, such as CDK4/6, mTOR and PI3K inhibitors [14].

The aim of this study was to address endocrine therapy resistance development with focus on early events driving the resistance process. We identified activating transcription factor 3 (ATF3), a transcription factor of the ATF/cAMP responsive element binding (CREB) family, as a novel mediator of endocrine resistance. Being part of a family of stress-response factors, upon induction by stress signals, its expression is rapidly increased to alter cellular processes that are relevant also in cancer progression [15,16,17]. To our knowledge, this study is the first reporting an involvement of ATF3 in endocrine resistance in breast cancer, and our data propose it as a central player in the early events of resistance development.

## 2. Results

### 2.1. Time-Resolved Profiling of Resistance Development Reveals ATF3 Upregulation

To investigate transcriptional dynamics in the process of acquisition of endocrine therapy resistance we chronically treated a luminal A cell line T47D for one year with either 100 nM 4-hydroxytamoxifen (TAM) or estrogen (E2) deprivation until we obtained T47D-tamoxifen-resistant (T47D-T) and T47D-long-term estrogen deprived (T47D-L) (Appendix A). Cells treated with TAM were resistant to the applied dose already after five months of treatment and showed increased resistance even to 10-fold higher TAM concentrations after seven months (Appendix A). At the same time, TAM-treated cells were able to sustain a steady growth, while the E2-deprived cells remained in a non-proliferative state until later timepoints (Appendix A). In agreement with the literature [18], E2-deprived cells showed a significant increase in invasion compared to treatment-naïve T47D, while no such difference was observed for TAM-treated cells (Appendix A).

To unravel novel drivers of the resistance process we performed time-resolved RNA-seq profiling of cells at one, two, five, and seven months during resistance development and we selected genes differentially upregulated in the early treatment time points, namely at one and two months (Figure 1a). This selection included a total of 768 and 761 genes for TAM-treated and E2-deprived cells, respectively, of which 282 were shared between the treatment groups (Appendix A). Pathway analysis of these overlapping genes revealed a significant enrichment in the MAPK signaling pathway, a pathway known to be de-regulated in endocrine resistance (Figure 1b, blue bars) [19,20]. To identify possible transcriptional drivers of the resistance process we used the C3 MSigDB transcription factors (TFs) motif collection to further investigate regulating genes of the MAPK signaling pathway. This identified several ATF3 motifs as significantly enriched in the deregulated genes. Among the TFs with motifs enriched in the regulatory regions of those genes, only *ATF3* was upregulated itself (Figure 1a, Appendix A), rendering this a good candidate for further analysis. Next, we performed targeted proteomic screening with reverse phase protein array (RPPA) and confirmed that central active phosphoproteins involved in the MAPK pathway as well as its deeply interconnected PI3K/AKT pathway were upregulated in resistant cells compared to their sensitive counterparts (Figure 1c). A similar pathway activation profile was detected in MCF7, a second resistant luminal A cell line [18] that we analyzed to exclude cell line-specific effects (Figure 1c). Central phosphorylated kinases responsible for signal transduction like AKT, MEK and p38 showed a similar upregulation in both cell lines. Increased activities of other proteins were rather treatment-specific (e.g., p-JNK in TAM-resistant cells) or cell line-specific (e.g., p-P44/42 in T47D). Notably, *ATF3* has been found to be among the top depleted genes in a CRISPR screen with MCF7 cells treated with TAM or Fulvestrant [21], thus confirming its role in endocrine therapy response (Figure 2a). We then tested if *ATF3* was already induced at timepoints earlier than one month of treatment with endocrine therapies. Indeed, in both cell lines, *ATF3* was upregulated as early as 1 week after treatment administration (Figure 1d) Additionally, *ATF3* expression was found elevated at the gene expression level in both the endocrine resistant models of T47D and MCF7 (Appendix A). Upregulation of ATF3 in the resistant cells was detected also at the protein level (Figure 1e, Appendix A). Since ATF3 is a member of the activating transcription factor family, which includes seven members with shared functions, we next investigated the expression of *ATF1-7* in the T47D RNA-seq data (Appendix A). Notably, *ATF3* was the only member of the family being upregulated, indicating a specific function of ATF3 in this context.

### 2.2. ATF3 Knockout Affects Cell Proliferation, Cell Cycle, Apoptosis, and Invasion

To investigate the role of ATF3 in resistance-related cellular processes we applied CRISPR-Cas9 technology to knockout ATF3 in sensitive MCF7. Two homozygous biallelic knockout clones defined as KO1 and KO2, representing two different sgRNAs, were derived from single cells and showed an insertion of 1 bp and a deletion of 17 bp respectively (Appendix A). Efficacy of the knockout was verified at the protein level: while WT cells showed *ATF3* expression after anisomycin treatment [22], no protein was detectable in the two knockout clones (Appendix A).

No drastic effects on proliferation were observed when WT and KO cells were grown in presence of E2 (Figure 2a). In the presence of TAM and under estrogen deprivation, however, the two *ATF3* knockout clones displayed a significantly slower proliferation rate, with KO1 showing the strongest effect (Figure 2a). Under the same conditions, the two knockout clones showed a reduction in cellular viability after estrogen deprivation or with TAM treatment (Appendix A). Similar effects were detected in a siRNA knockdown approach in both MC7 and T47D cell lines (Appendix A).

To understand if the effect of ATF3 knockout on proliferation and viability under treatment also affected other cellular processes, we next tested the cells for their cell cycle distribution and apoptosis rates. While no significant differences in the cell cycle distribution were detected in presence of E2, knockout clones showed a striking reduction in the percentage of cells entering S phase compared to their WT counterpart, both under TAM and -E2 treatment (Figure 2b). Interestingly, ATF3 knockout also affected the percentage of apoptotic cells under treatment: both ATF3 KO clones showed a significant increase in the percentage of early apoptotic cells upon TAM treatment and E2 deprivation (Figure 2c). The effect of ATF3 depletion on cell cycle and apoptosis was also assessed with RNAi, obtaining similar results in both MCF7 and T47D (Appendix A).

ATF3 has been implicated in the regulation of cellular invasion in several cancer entities [23,24]. To assess if the knockout altered the invasion capabilities, the cells were tested in a transwell matrigel assay. MCF7 are not highly invading cells and in baseline conditions they have limited motility. Indeed, without stimulation all the cells showed low invading abilities with KO2 displaying a significantly lower number of invading cells (Figure 2d). However, upon TFGβ1 stimulation, the WT cells showed the expected increase in invasion, while the two knockouts showed similar invasion rates as in the unstimulated condition (Figure 2d). These data suggest a potential role of ATF3 in TGFβ1-induced invasion in breast cancer.

Altogether, these data demonstrate that ATF3 plays a crucial role in cellular response to treatment and that the lack of ATF3 increases the sensitivity of cells to endocrine treatments affecting both cell cycle progression and the apoptosis rate.

### 2.3. ATF3 Overexpression Induces Resistance to Endocrine Therapy

Since *ATF3* knockout increased the sensitivity of the cells to TAM treatment and E2 deprivation, we next tested if its overexpression could render the cells more resistant. To this end, we used a lentiviral approach to obtain WT MCF7 and T47D cells stably overexpressing ATF3. The overexpression was indeed evident both at the mRNA and protein levels in both cell lines compared to the empty vector control (Appendix A).

While proliferation in MCF7 cells was not affected by overexpression of ATF3 in media containing E2, TAM treatment affected the growth rate of ATF3 overexpressing cells significantly less than the control cells (Figure 3a). Viability measurements confirmed this difference, with ATF3 overexpressing cells showing a significantly increased viability under TAM conditions (Appendix A). In contrast to MCF7, T47D cells overexpressing ATF3 displayed already higher proliferation than the empty control in presence of E2 (Appendix A). However, the difference increased when the cells were kept in the presence of TAM or without E2, indicating a more resistant phenotype of the T47D cells overexpressing ATF3 (Appendix A), also supported by the viability data (Appendix A).

Next, we tested the effect of ATF3 overexpression on cell cycle and apoptosis. No significant difference was observed in the number of MCF7-cells entering S phase between the empty control and the overexpressing cells, when assessed in E2 containing media. Under TAM treatment and E2 deprivation however, ATF3 overexpressing cells showed an increase in S phase, demonstrating less sensitivity to the treatments (Figure 3b). Additionally, while no significant difference in the apoptosis rate was observed between the empty and the overexpressing cells with or without E2, the percentage of early apoptotic cells induced upon TAM treatment was significantly lower in the ATF3 overexpressing cells (Figure 3c). While a similar trend was observed for the E2-deprived condition the effect did not reach significance. Similar to MCF7, in T47D, no difference was observed in the percentage of cycling cells between empty and ATF3 overexpressing cells when kept in E2 media. However, ATF3 overexpression was able to prevent the cell cycle arrest induced by TAM, while no significant difference was observed under E2 deprivation (Appendix A). At the same time, in T47D, ATF3 overexpression was able to significantly reduce TAM-induced early apoptosis compared to the empty control. A decrease in the early apoptotic cells, even if not statistically significant, was detected also in ATF3 overexpressing cells deprived of E2 (Appendix A).

Since ATF3 knockout affected the invasion capabilities of MCF7, we then tested if overexpression of ATF3 was able to induce the opposite phenotype. While in unstimulated conditions ATF3 overexpressing MCF7 cells displayed a similar number of invading cells as the empty control, upon stimulation the increase in invasion was enhanced in the overexpressing cells (Figure 3d). Both T47D cell lines (empty control and ATF3 overexpressing) were not able to invade through matrigel (data not shown). Overall, these data demonstrate that ATF3 overexpression is able to induce a resistant phenotype in sensitive cells.

### 2.4. ATF3 Modulation Affects Downstream Pathway Activities

The RNA-seq data had revealed that the cells rewire cellular processes during resistance development, mainly affecting genes involved in MAPK signaling (Figure 1b). Additionally, resistant cells showed higher levels of many phospho-proteins involved in this pathway (Figure 1c), indicating active signal transduction. To investigate if the knockout of ATF3 would directly affect the pathway activation profile of the cells under treatment, we applied the RPPA approach again. Acute treatment of WT MCF7 with endocrine therapies induced the upregulation of several phospho-proteins in the AKT and MAPK pathways involved in stress-response, including central kinases as AKT, MEK, JNK1 and p38 (Appendix A). However, we saw no increase in the phosphorylation of the aforementioned central kinases in the knockout, indicating the inability of these cells to activate these pathways upon treatment/stress. Concordantly, eiF4B phosphorylation is decreased upon treatment in the ATF3 knockout cells (Appendix A), which is in line with the increased sensitivity of the knockout to treatment (Figure 2). Considering the drastic differences in pathway activation profiles upon treatment of the knockout cells, these results suggest that ATF3 knockout indeed affects the regulation of several branches of the MAPK and PI3K/AKT pathways in response to endocrine treatment and the lack of ATF3 prevents the cells from augmenting the expression and activities of central proteins involved in these pathways. As ATF3 overexpression induced the opposite phenotypic effects compared to the knockouts, i.e., increasing the resistance of sensitive cells to treatment, RPPA analysis was used also to compare the ATF3 overexpression to the empty vector control. Even though the effect on pathway alteration was not as strong as for the knockout, consistent changes in central proteins were detected. This is particularly evident after one week of treatment with E2 deprivation, which causes an increase in p38, ERK, and AKT phosphorylation compared to the empty vector control (Appendix A). In an opposite trend compared to the knockouts, also eiF4B phosphorylation is induced in the overexpressing cells upon treatment, in agreement with their more proliferative phenotype under treatment (Figure 3).

### 2.5. ATF3 Knockout Affects Tumor Growth In Vivo

ATF3 knockout had a striking effect on proliferation, cell cycle progression and apoptosis in vitro (Figure 2). To assess if these effects were reproducible in an in vivo xenograft model, E2-pretreated mice were orthotopically injected with the ATF3 knockout clones and their WT counterpart. After randomization, treatment was continued with E2or TAM pellets (Figure 4a). In the presence of E2, all mice in the three groups had to be sacrificed due to tumor progression, with the two ATF3 knockouts reaching the humane endpoints slower than the WT (Figure 4b).

Under TAM treatment, similarly to what has been seen in vitro, both ATF3 KO clones showed slower growth. Astonishingly, all the mice injected with ATF3 KO1 displayed no growth after the initial engraftment when treated with TAM, with one mouse even showing complete remission (Figure 4b). A growth delay was detected also in the KO2, where two mice were sacrificed as in the WT group, but much later in time (Figure 4b). Additionally, among the mice still alive at the end of the experiments, the WT ones presented the biggest tumors, with the two KO having significantly smaller masses (Figure 4c). These data confirm the in vitro data and support the role of ATF3 in the regulation of resistance to endocrine therapy.

### 2.6. ATF3 Increases upon Endocrine Administration in Patients

To evaluate the clinical relevance of the in vitro and in vivo findings, we investigated *ATF3* levels in publicly available datasets with gene expression data from patients treated with endocrine therapy. *ATF3* was found significantly upregulated after therapy administration in five out of six available datasets (Figure 5). In all the datasets *ATF3* showed a significant increase after few months of therapy administration (>90 days), but notably in GSE80077, ATF3 was already upregulated after 14 days of treatment. As this is the only datasets including premenopausal patients treated with TAM, we assessed if this was the cause driving this difference. However, both TAM and LET are inducing ATF3 expression in GSE80077 after 14 days, indicating that this difference is not related to TAM specifically (Appendix A)

Strikingly, in GSE111563 dataset, *ATF3* was found to be the top differentially expressed gene between the three time points (Appendix A). Notably, many of the other top differentially expressed genes are predicted targets of ATF3 (*EGR1*, *DUSP1*, *FOSB*, *JUN*), or upregulated genes in the early time points cell line model (*DUSP1*, *FOS*, *JUN, GEM*), corroborating the relevance of ATF3 in the regulation of the response to endocrine therapy. Interestingly the early upregulated gene set (Appendix A) showed an enrichment in the treated biopsies compared to the pre-treatment biopsies in patients from GSE111563 (Appendix A). This overlap is a strong confirmation of the relevance of the cell line model used to recapitulate effectively in vitro the clinical gene expression profiles of treated tumors. Overall, these data confirm the increase in expression of ATF3 upon therapy administration and support the relevance of ATF3 in the clinical setting.

## 3. Methods

### 3.1. Cell Culture

The well characterized luminal A [25] breast cancer cell line models MCF7 and T47D were obtained from ATCC (LGC Standards GmbH, Wesel, Germany). Cell lines were cultured in Dulbecco’s Modified Eagle Medium (DMEM) supplemented with 10% FCS, 50 units/mL penicillin and 50 µg/mL streptomycin sulfate (Invitrogen AG, Carlsbad, CA, USA), and 10^−8^ M 17-ß-estradiol (E2, Sigma-Aldrich, Saint-Louis, MI, USA) and were kept at 37 °C with 5% CO_2_ in a humidified atmosphere. The more potent active metabolite 4-hydroxytamoxifen was used instead of tamoxifen. Tamoxifen resistant cells were maintained in 100 nM 4-hydroxytamoxifen (TAM, Sigma-Aldrich) and E2-deprived cells were kept in DMEM (*w*/*o* phenol red) supplemented with 10% charcoal stripped FCS. T47D-T (tamoxifen resistant) and T47D-L (long-term estrogen deprived) cell lines were generated by chronically treating sensitive cells with TAM containing or E2 deprived media for one year in two independent biological replicates each. MCF7-T (tamoxifen resistant) and MCF7-L (long-term estrogen deprived) were kindly provided by Dr. Luca Magnani form Imperial College London (ICL), London, UK. All cell lines were regularly authenticated (Multiplexion GmbH Heidelberg, Germany) as well as tested for potential mycoplasma contamination. For transient knockdowns cells were transfected the day after seeding with RNAiMax^®^ (Invitrogen AG, Carlsbad, CA, USA) according to manufacturer’s instructions. siRNAs (siTOOLs Biotech, Planegg, Germany) were used at a final concentration of 2 nM in P/S free media.

### 3.2. Generation of Stable Cell Lines

Virally transduced stable cell lines were generated in the stable isogenic cell line core facility at the DKFZ. Briefly, HEK293FT cells were co-transfected with the lentiviral constructs (pLX-304 vector with/without ATF3 ORF) and second-generation viral packaging plasmids VSV.G (Addgene #14888) and psPAX2 (Addgene #12260). 48 h after transfection, virus containing supernatant was removed and MCF7 and T47D cells were transduced with lentiviral particles at 70% confluency in the presence of 10 μg/mL polybrene. Transduced cells were selected with blasticidin (Thermo Fisher Scientific, Waltham, MA, USA).

ATF3 knockout clones were generated with CRISPR/Cas9 technology using two sgRNAs (KO1 -AAAGUGCCGAAACAAGAAGA, KO2-AGAAGGCACUCACUUUCUGC, (Sinthego, Redwood City, CA, USA) targeting exon 3 of the ATF3 gene. Cells were transfected with Lipofectamin CRISPRMAX Cas9 transfection reagent (Thermo Fischer Scientific) according to the manufacturer’s instructions. Single-clone derived colonies were screened for successful editing both at the DNA level with Sanger sequencing (Eurofins Genomics, Ebersberg, Germany) and at the protein level with Western Blot. No viable ATF3 knockout clones could be obtained from T47D.

### 3.3. RNA Isolation and qRT-PCR

RNA was isolated and purified using the RNeasy Mini Kit (Qiagen, Hilden, Germany) according to manufacturer’s instructions. RNA concentration was measured with NanoDrop-ND 1000 and cDNA was synthesized using the RevertAid H Minus First Strand Reverse Transcription Kit (Thermo Fischer Scientific). Quantitative Reverse Transcription-PCR (qRT-PCR) for target genes was performed using primaQUANT real-time PCR Master Mix (Steinbrenner Laborsysteme, Wiesenbach, Germany) using QuantStudio™ 3 or 5 Real-Time PCR Systems (Applied Biosystems) and Universal Probe Library probes, UPL (Roche). Data were analyzed using the SDS or QuantStudio software with the ΔΔCt method. The Ct values were normalized to housekeeping gene *ACTB*. Primers: ATF3 forward: TTTGCCATCCAGAACAAGC; reverse: CATCTTCTTCAGGGGCTACCT. ACTB forward: ATTGGCAATGAGCGGTTC; reverse: GGATGCCACAGGACTCCA.

### 3.4. RNA Sequencing

RNA sequencing (RNA-seq) was performed using Illumina HiSeq 4000 paired-end 100 base pair sequencing in the Genomics and Proteomics Core Facility at the DKFZ. Raw sequencing data were subjected to quality control using FastQC and trimmed using Trimmomatic (v0.32) [26]. Reads were aligned to the human reference genome (hg38) using HISAT2 (v2.0.4) [27]. Gene level counts were obtained using HTSeq (v0.6.1) [28] and the human GTF from Ensembl (release 85). Statistical analysis was performed using the edgeR [29] and limma [30] R/Bioconductor packages. Genes with more than five counts in one or more samples were retained. Count data were transformed to log2-counts per million (logCPM) with a prior count of three and normalized by applying the trimmed mean of M-values method. Gene-wise linear models were fitted with coefficients for each combination of treatment (+E2, +TAM, -E2) and time point and a coefficient to correct for systematic differences between the two biological replicates. For both E2 deprived and TAM treated samples, contrasts were made between each individual time point t and the WT cell line, that is, (-E2_t_)–untreated_0_ and (+TAM_t_)–untreated_0_, respectively. Differential expression was assessed using empirical Bayes moderated statistics with an intensity-dependent trend fitted to the prior variances [31]. Resulting *p*-values were corrected for multiple testing using the Benjamini–Hochberg false discovery rate (FDR). Genes that changed in the early treatment time points (one or two months) compared to E2 were selected based on their moderated F-statistics and corresponding FDR < 0.1 (-E2) or FDR < 0.25 (+TAM). Additional gene annotation was retrieved from Ensembl (release 91) using the biomaRt R/Bioconductor package. Gene set enrichment analysis was performed using a one-sided hypergeometric test with the kegga function (*limma* package) using all genes identified in the RNA-seq experiment as a background universe. Genes sets investigated were either KEGG pathways or motif genesets from the C3 collection of the Molecular Signatures Database (MSigDB, v6.1) [32].

### 3.5. Western Blot

Cells were lysed with RIPA lysis buffer (Thermo Fisher Scientific) supplemented with protease inhibitor Complete Mini and phosphatase inhibitor PhosSTOP (Roche). Protein concentrations were determined using the BCA Protein Assay Kit (Thermo Fisher Scientific) according to manufacturer’s instructions. Lysates were mixed with 4x RotiLoad and heated to 95 °C for 5 min. Mini-PROTEAN TGX Precast Gels were loaded with the lysates and electrophoresis was performed at 145V for 60min. Proteins were then transferred to a PVDF membrane using the Trans-BlotR Turbo Transfer System accordance to manufacturer’s instruction. Membranes were blocked with Rockland blocking buffer and subsequently incubated with target-specific primary antibodies ON at 4 °C. Alexa Flour 680 and 800 conjugated secondary antibodies were used for visualization, scanning the membranes with the Odyssey Infrared Imaging System (LI-COR Biosciences, Bad Homburg, Germany). Bands intensities were quantified using Image-J software (NIH).

### 3.6. Reverse Phase Protein Array

RPPA experiments were performed as previously described [33,34,35]. Briefly, cell lysates from three biological replicates for each condition were spotted in nitrocellulose-coated glass slides (Oncyte Avid, Grace-Biolabs, Bend, OR, USA) in technical triplicates using the Aushon 2470 contact printer equipped with 185 µm solid pins. All primary antibodies used were previously validated using Western bloting to prove specificity. Signal intensities of spots were quantified using GenePixPro 7.0 (Molecular Devices) and RPPA raw data preprocessing and quality control were performed using the *RPPanalyzer* R-package [36]. Intensity values were log2 transformed and plotted using Morpheus software (https://software.broadinstitute.org/morpheus). Log2 transformed values are listed in Appendix A and antibodies used in the study are listed in Appendix A.

### 3.7. Cell Viability and Proliferation

Cells were seeded in black clear-bottomed 96 well plates, and DNA was stained with fluorescent intercalating dye Hoechst-33258. Plates were imaged with the IXM XLS microscope (Molecular Devices), and all nuclei within a certain size and intensity were detected and counted with the Molecular Devices Software. Cell viability was measured with CelltiterGlo luminescent assay (Promega) according to manufacturer’s instructions. Luminescence was determined using the Glomax Explorer Plate Reader.

### 3.8. Invasion Assay

Cells were seeded in the upper chamber of BioCoat Matrigel Invasion Chambers (Corning, Kaiserslautern, Germany) in DMEM without serum. DMEM supplemented with 20% FBS was used as chemoattractant in the lower chamber. After 72 h cells in the lower membrane were by fixed with 4% PFA and stained with crystal violet. For quantification, the transwells were either eluted with 10% acetic acid and quantified with the Glomax Explorer Plate Reader measuring the absorbance at 590 nm or imaged with the LSM 800 microscope (Zeiss). Images were exported and analyzed using Image-J software. T47D cells were not able to invade through the invasion chambers.

### 3.9. Analysis of Apoptosis and Cell Cycle

Apoptosis rate was assessed with the Annexin/PI kit (BD Bioscences) and analyzed with Flow Cytometry (FACS) according to manufacturer’s instructions. Cell cycle distribution was assessed with Bromodeoxyuridin (BrdU) and 7-Aminoactinomycin D (7-AAD) staining. Cells were starved with DMEM without FCS for 20 h to synchronize the cells. The cell cycle block in G0 was released by incubation of cells in full growth media supplemented with 10 µM BrdU. After 2 h, cells were permeabilized with the Perm/Wash buffer and fixed with the Cytofix/Cytoperm buffer for 20 min at RT. Cells were incubated with DNase for 1 h at 37 °C to expose the BrdU. FITC-anti BrdU antibody was used for detection. Samples were analyzed with Flow Cytometer FACS Calibur (BD Biosciences). Analysis was performed with FlowJo v10 software.

### 3.10. In Vivo Experiments

Animal experiments were performed in the DKFZ animal facility (animal experiment licensed under G272-16) using eight-week-old female Nod Scid Gamma mice recruited from the in-house breeding cohort. All applied pellets were from Innovative Research of America. Twenty-one-day release estrogen pellets (0.25 mg/pellet) were subcutaneously implanted by trochar into the neck region of all the mice to promote tumor engraftment. After seven days, the mice were randomized and tumor cells were injected in the mammary fat pad (MFP) with 2.5 million cells in 30 µL of PBS: matrigel (1:1, *v*/*v*). Fourteen days thereafter, after randomization, the treatment with 60-day-release pellets (either E2 or TAM) was started and then replaced with new pellets after 60 days. Mice were sacrificed when tumors had reached 1 cm of diameter in any dimension, if they presented health problems, had lost more than 20% of their maximal weight, or if they had reached the planned end point of the observation period (120 days of treatment). Six mice were used for each cohort. Two mice treated with E2 had to be sacrificed because of health issues and their data were excluded from further analysis.

### 3.11. Clinical Datasets Analysis

Six GEO [37] datasets (GSE80077 [38], GSE10281 [39], GSE20181 [40], GSE59515 [41], GSE55374 [42], GSE111563 [43]) containing data from ER+ patients preoperatively treated with neoadjuvant endocrine therapy were included in the analysis. GSE80077, GSE10281 and GSE20181 data were obtained with the Affymetrix Human Genome U133A 2.0 Array (ATF3 probe: 202672_s_at) GSE59515, GSE55374 and GSE111563 data were obtained with the Illumina HumanHT-12 V4.0 expression beadchip (ATF3 probe: ILMN_2374865). GSE80077 contains both pre- and post-menopausal patients treated with 40mg tamoxifen/day or 2.5 mg letrozole/day respectively. All the other datasets include only post-menopausal patients treated with 2.5 mg letrozole/day.

### 3.12. Statistical Analysis and Graphical Illustration

Unless otherwise mentioned, data are presented as the mean of three biological replicates ± SEM. Statistical analyses were performed by unpaired two-tailed Student’s *t*-test using GraphPad Prism Software. *p*-values < 0.05 were considered statistically significant and *p*-values < 0.05, < 0.01 and < 0.001 are indicated with one, two and three asterisks, respectively. All graphs were generated using the GraphPad Prism Software or R and illustrated via Inkscape v 0.91. Graphical abstract details were obtained from Servier Medical Art (https://smart.servier.com/).

## 4. Discussion

Therapy resistance is an urgent clinical problem that affects up to 40% of luminal breast cancer patients treated with endocrine therapy [10,11,13]. Even though several mechanistic studies investigated the reasons behind resistance development and novel second-line therapies are being used in recurrent patients (e.g., CDK4/6 inhibitors, mTOR inhibitors, PI3K inhibitors) [44], the prognosis for relapsing patients has generally remained poor, and more efforts are required to identify new targetable genes and pathways involved in this process. Studies that aim to recapitulate the resistance development in vitro commonly focus on resistant cell line models and compare these to their sensitive counterpart. This approach has revealed numerous key alterations involved in resistance development and has led to the development of new therapeutic approaches. However, there is a lack of knowledge regarding genes responsible for the survival of the cells to the acute cytotoxic effect of the drugs and the rewiring of their molecular features. From a therapeutic perspective, it is essential to understand the drivers of the resistance phenomena, rather than the downstream alterations, to be able to propose new first-line targets that can help to delay the resistance development or prevent it. In this view a deep understanding of the early phases of the treatment may help to pinpoint causes and effects, and how they are connected. Using a longitudinal screening during the development of endocrine resistance, we identified ATF3 as a novel driver of the resistance process through the rewiring of the MAPK pathway. The physiological role of ATF3 varies in different cells, but it has a well-characterized role in the regulation of the stress caused by DNA-damage repair and cell cycle progression [45].

The role of ATF3 in the regulation of transcriptional activity depends on the binding partner and context. As an example, binding of ATF3 to JUN or JUND usually represses transcription, while its dimerization with JUNB can have both active and repressive roles [46]. Due to its potential dual role, ATF3 has been found to have different effects on cancer progression. Elevated ATF3 expression has been correlated with poor prognosis in prostate cancer and has been associated with increased proliferation and metastasis formation [47]. Similarly, ATF3 was reported to promote cell invasion and to contribute to tumor spreading in colon cancer [48]. Additionally, ATF3 knockdown has been proved to impair cellular growth and viability in Hodgkin lymphoma, glioblastoma, and lung cancer [24,49,50]. On the other hand, several studies have reported oncosuppressive roles of ATF3. Its overexpression in liver cancer and hepatocellular carcinoma cells reduced proliferation and motility while apoptosis was increased [51,52]. Other reports showed ATF3 as inhibitor of invasion and migration both in colorectal cancer and ovarian cancer cells [53,54,55]. It thus seems that the role of ATF3 in cancer progression is highly context dependent and tumor-type specific.

In line with our findings, ATF3 has been mostly characterized as an oncogene in breast cancer. The *ATF3* gene maps to chromosome 1q32.3 in the q1 amplicon. This locus is amplified in around 53% of all breast cancers making this the most amplified chromosomal region in breast cancer [56]. Indeed overexpression of ATF3 was able to induce spontaneous lesions in the mammary glands of multiparous mice therefore acting as a potent oncogene [57]. Further studies found that its oncogenic role is mediated by the upregulation of the WNT/β-catenin pathway, with the induced tumors being transcriptionally and phenotypically similar to breast tumors induced by WNT pathway overactivation [58]. Another report showed that ATF3 overexpression is able to promote cancer-initiating features in immortalized mammary epithelial MCF10A cells via the TGFβ pathway [59]. It is therefore established that ATF3 can rewire the mammary gland processes by affecting several downstream pathways involved in tumor initiation and progression. Additionally, elevated expression of ATF3 has been correlated with poor overall survival in breast cancer, even though in a cohort skewed towards the lobular carcinoma subtype [60]. ATF3 has been investigated also in the context of chemotherapy and radiotherapy response, surprisingly with opposite roles. While being upregulated upon both treatments, in chemotherapy it has been described as a mediator of cytotoxicity, whereas regulating resistance to treatment with radiotherapy [61,62]. Notably, higher ATF3 levels correlated with improved overall survival in response to chemotherapy [62]. On the other hand, ATF3 overexpression was able to enhance radioresistance both in vitro, promoting cell cycle progression and preventing apoptosis, and in vivo [61]. Our findings, with ATF3 being among the upregulated genes in response to endocrine therapy and promoting resistance development, are in line with the latter. Notably, in our system ATF3 was also a putative regulator of several genes that shared with ATF3 the particular profile of being upregulated after short-time therapy administration due to the presence of ATF3 motifs in their promoters. Several of these genes have been reported to be induced by endocrine therapy administration or related to resistance, as *DUSP10, FGF12*, and *PRICKLE2* [63,64]. Another striking example is the AP-1 complex which is well characterized as a mediator of endocrine resistance development [65,66]. Indeed, both immediate early stress-response genes *FOS* and *JUN* were upregulated upon TAM treatment and E2 deprivation. These are not only binding partners of ATF3 but are also ATF3 target genes. Additionally, it has been shown by ChIP-seq that ATF3 not only localizes to ATF3 TF/CRE motifs (5′-TGACGTCA-3′) but also the AP-1 binding motif (5′-TGASTCA-3′, S = C/G) [67] therefore indicating that AP-1 and ATF3 could act together in promoting resistance development.

ATF3 has been also described as a downstream effector of the MAPK pathway through several distinct branches and mechanisms in different contexts. First, *ATF3* promoter has been shown to be activated by the binding of ATF2 and c-jun, two downstream effectors of MAPK signaling [68]. ATF3 upregulation under stress was proven to be mediated exclusively by the p38 signaling pathway in HeLa cells, while through ERK, SAPK and p38 in colorectal cancer [22,69]. Other reports showed that its activation is mediated through ERK and p38α in myocytes or by ERK/JNK but not by p38 in rat brains [70,71]. Again, these findings demonstrate that the regulation of ATF3 is highly cell and context specific. In breast cancer and particularly in resistance to treatments, ATF3 expression has been described as mediated by pAKT in radioresistance, mostly by JNK pathway in chemoresistance [61,62]. Therefore, while the mechanisms of ATF3 induction through MAPK and AKT have been extensively described, the downstream activity of ATF3 to propagate the signal is not well studied. Our targeted proteomic approach with RPPA revealed a large difference in pathway activation profiles between WT cells, ATF3 knockout and overexpressing cells. While in the WT cells endocrine treatments induced activation in several central phospho-proteins as AKT, p38, MEK1, JNK, and eiF4B, the knockout clones displayed a generally less active MAPK and PI3K/AKT pathways. This suggest a central role of ATF3 in regulating MAPK and PI3K/AKT signaling pathways under therapy and that the lack of ATF3 prevents activation of these pathways as well as of the resistance development process.

Our data also validated the relevance of ATF3 in vivo in a xenograft mouse model. ATF3 knockouts displayed slower growth in vivo under TAM treatment, confirming the in vitro results. Differently from their behavior in vitro, the knockouts also showed a slower growth without any treatment administration, indicating a possible role of ATF3 in controlling the baseline proliferation rate of the cells.

Finally, we also showed that ATF3 was induced upon endocrine therapy in patients’ biopsies in most of the datasets analyzed, adding clinical relevance to the findings.

## 5. Conclusions

While we demonstrated the role of ATF3 in endocrine resistance development and its correlation with clinical data, further studies are needed to clarify the exact mechanism by which ATF3 can regulate MAPK/AKT pathways and promote the survival of the cells under stress. Understanding the downstream effectors of ATF3 could help to identify targetable gene products for an effective combinatorial treatment in the clinic. Several MAPK/AKT inhibitors and antibodies are currently tested in the clinic as second or third line treatment for endocrine therapy resistant patients [72]. However, these trials are assessing treatment efficacy in a resistant setting instead of combining these therapies to endocrine standard treatment as first line in selected patients. We suggest that the timing of treatment is essential for therapeutic efficacy: intervening with these inhibitors during the tumor’s adaptation to the treatment could result in the blockage of the pathways involved in resistance development and inhibit disease recurrence.

## Figures and Tables

**Figure 1 cancers-12-02918-f001:**
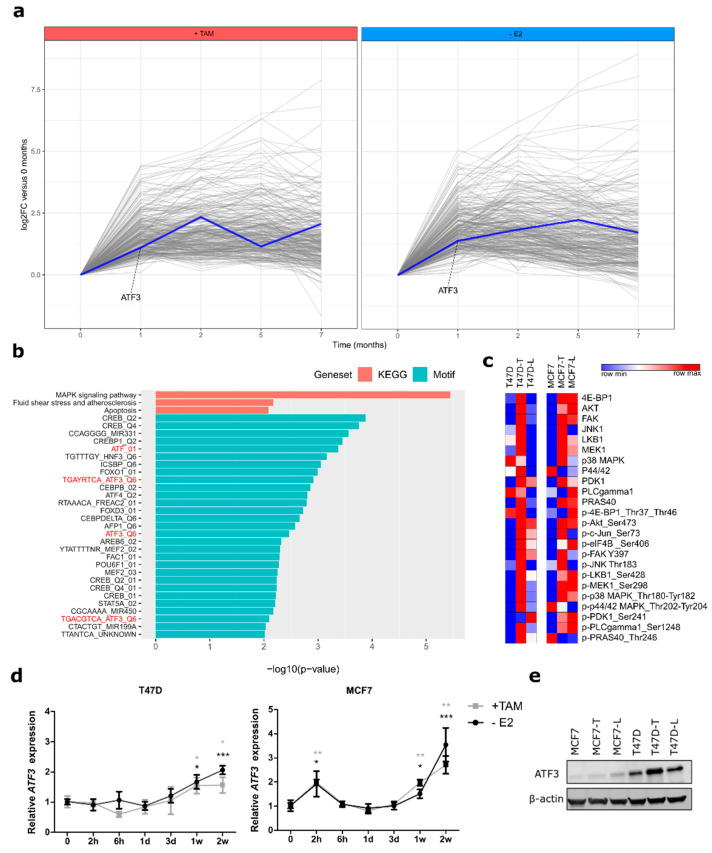
Time-resolved profiling identifies activating transcription factor 3 (ATF3) upregulation in early stages of resistance development. (**a**) Differentially upregulated genes after one or two months of treatment of T47D with 4-hydroxytamoxifen (TAM) (left panel) or E2 deprivation (right panel). *ATF3* profile is highlighted in blue. RNA-seq values are represented as relative Log2 Fold Change compared to parental cells (**b**) Red: pathway enrichment analysis on the 282 commonly upregulated genes between TAM-treated and E2-deprived cells using KEGG (Kyoto Encyclopedia of Genes and Genomes) pathways. Blue: putative transcription factors binding analysis on the 282 commonly upregulated genes between TAM-treated and E2-deprived cells using Molecular Signature Database C3 motifs collection. (**c**) Heatmaps of 24 total proteins and phospho-proteins involved in MAPK and PI3K/AKT signaling pathways in T47D and MCF7 sensitive and resistant cells. Log2 normalized signal intensities for each protein are plotted and color-coding refers to relative intensities in each row independently. (**d**) Relative *ATF3* mRNA levels determined by qRT-PCR in T47D and MCF7 treated for two weeks with either TAM or E2 deprivation. (**e**) ATF3 protein levels in sensitive and resistant T47D and MCF7 determined by WB after 2h anisomycin stimulation. β-actin levels are used as loading control. qRT-PCR data are normalized to *ACTB* and then to untreated cells and represented as mean + SEM, n = 3. *** *p*-value < 0.001, ** *p*-value < 0.01, * *p*-value < 0.05.

**Figure 2 cancers-12-02918-f002:**
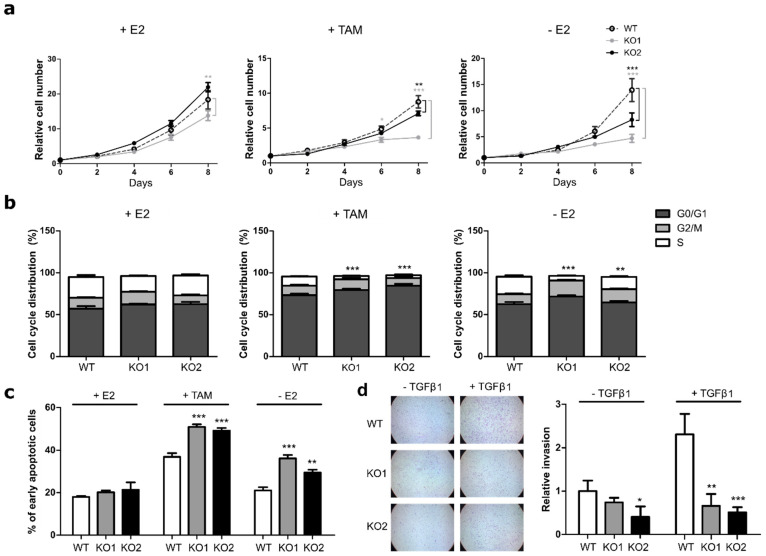
ATF3 knockout increases sensitivity to endocrine therapy. (**a**) Cell proliferation of MCF7 cells treated for eight days with E2, TAM, or deprived from E2 and measured at indicated time points with nuclei count in fluorescent microscopy. All values are normalized to a seeding control. (**b**) Cell cycle distribution of MCF7 cells treated for four days with E2, TAM or without E2. Plots represent the percentage of cells in the different cell cycle phases determined by BrdU/7AAD staining. Statistics performed on the S phases (white bars) (**c**) Measurement of apoptosis rate in MCF7 cells treated for four days with E2, TAM, and without E2. Plots represent the percentage of early apoptotic cells determined by Annexin V/PI staining. (**d**) Representative microscopy images of transwell invasion assay through Matrigel and relative quantification of the number of invading cells. Values are expressed as relative to the unstimulated WT control. Data are represented as mean ± SEM, *n* = 3. *** *p*-value < 0.001, ** *p*-value < 0.01, * *p*-value < 0.05.

**Figure 3 cancers-12-02918-f003:**
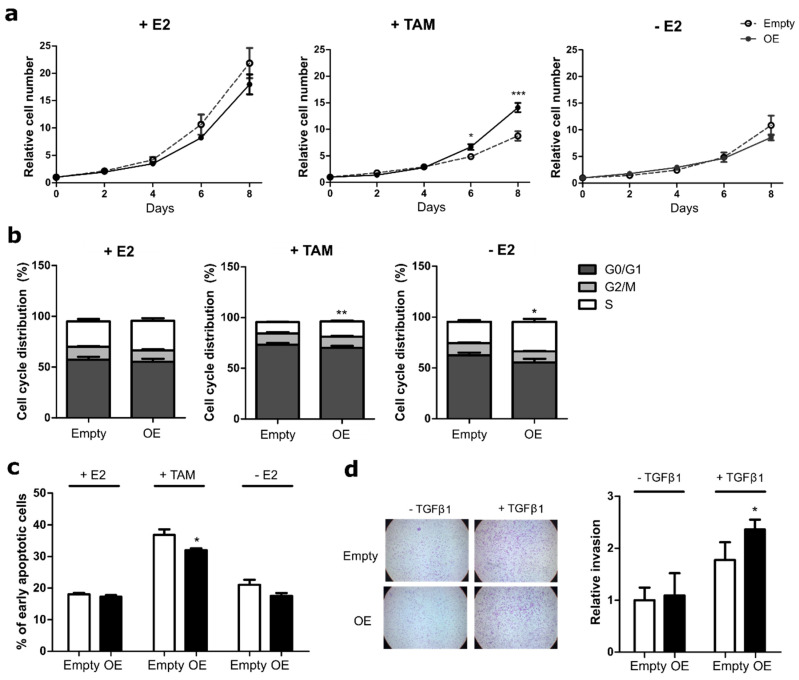
ATF3 overexpression reduces sensitivity to endocrine therapy in MCF7 cells. (**a**) Cell proliferation of MCF7 empty and ATF3 overexpressing (OE) cells treated for eight days with E2, TAM, or deprived from E2 and measured at indicated time points with nuclei count in fluorescent microscopy. All values are normalized to a seeding control. (**b**) Cell cycle distribution of MCF7 empty and ATF3 OE cells treated for four days with E2, TAM, or without E2. Plots represent the percentage of cells in the different cell cycle phases determined by BrdU/7AAD staining. Statistics performed on the S phases (white bars) (**c**) Measurement of apoptosis rate in MCF7 empty and ATF3 OE cells treated for four days with E2, TAM and without E2. Plots represent the percentage of early apoptotic cells determined by Annexin V/PI staining. (**d**) Representative microscopy images of transwell invasion assay through Matrigel and relative quantification of the number of invading cells. Values are expressed as relative to the unstimulated WT control. Data are represented as mean ± SEM, n = 3. *** *p*-value < 0.001, ** *p*-value < 0.01, * *p*-value < 0.05.

**Figure 4 cancers-12-02918-f004:**
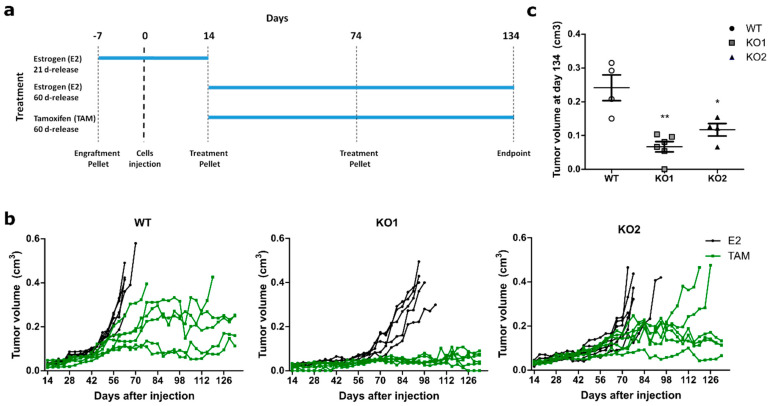
ATF3 knockout impairs in vivo tumor growth upon tamoxifen treatment. (**a**) Schematic representation of the animal experiment. (**b**) Tumor volume over time up to 120 days of treatment. Tumor growth curves are plotted for mice treated first with an E2 21-day-release pellet (0.25 mg) seven days prior to injection with MCF7 WT or knockout cell clones. 14 days thereafter continuous treatment started with 60-day-release pellets consisting either of E2 (0.72 mg) or TAM (5 mg) throughout the experiment. Each line represents a mouse (n = 6 for each cohort) and each dot represents a measurement. (**c**) Tumor volumes of TAM-treated mice still alive at the experimental endpoint. Final tumor values represented as mean ± SEM. ** *p*-value < 0.01, * *p*-value < 0.05.

**Figure 5 cancers-12-02918-f005:**
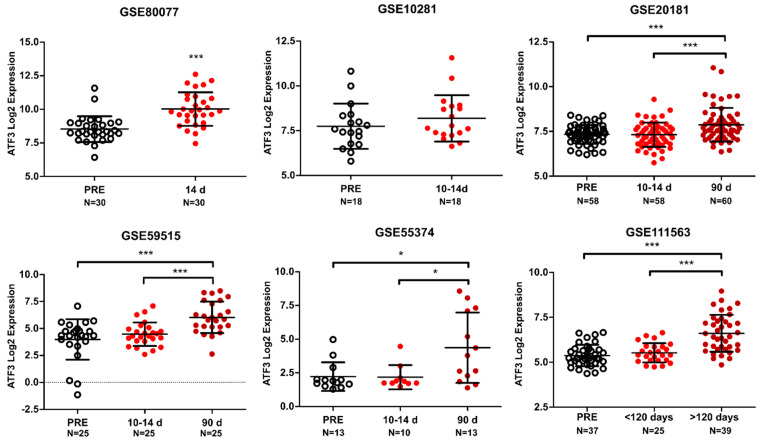
*ATF3* expression in patients treated with endocrine therapy. *ATF3* expression in six GEO datasets (GSE80077, GSE10281, GSE20181, GSE59515, GSE55374, GSE111563) comprised of matched tumors before (PRE) and after neoadjuvant endocrine therapy administration (in days). All values are represented as Log2 expression of the respective probe in individual patients ± SD. *** *p*-value < 0.001, * *p*-value < 0.05.

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
