# Peer review of "Time-Resolved Profiling Reveals ATF3 as a Novel Mediator of Endocrine Resistance in Breast Cancer"

_cancers, 2020, doi:10.3390/cancers12102918_

Round 1

Reviewer 1 Report

A very well written and interesting article, some isusse do need some further explanation/ correction.

In most part of the article you discuss results related to treatment of cells with TAM in the results this is translated in 100 nM tamoxifen but in the methods TAM seems to be 4-Hydroxytamoxifen the much more active metabolite of tamoxifen. Please clarify through the whole article what you have been using in the different experiments. If you have used 4-Hydroxytamoxifen, which is better in my view, you should explain this beter in the introduction why you choose to do so.

In the introduction the authors mention that 30-40 develop resistance to the treatment based upon two older studies please provide newer references (have a look at the ATLAS study). These older studies sometime still include patients before screening programmes were introduced or patients that received chemotherapy first and that would not be representative for your hypothesis.

Your hypothesis is that ATF3 is a central player in the early events of resistance development, so why do you then use cell lines that are obtained from metastic patients. Do these cell lines really reflect your target patient group? Is so please explain.

In the discussion you state that the role of ATF3 in cancer progression is highly context dependent and tumor-type specific. You refer to several articles on ATF3 in breast cancer and their several different roles have been described. Please add more comment on these different studies. Reference 47 is a particular bad one as it hardly represents a "normal" breast cancer population (the patient populations consists of 15 DCIS and 44 lobular carcinoma patients out of a total of 114.) Pleas add and discuss data from KM-plotter for tamoxifen treated patients only.  

In the conclusion the authors state that the level ATF3 in primary tumors does not correlate with disease-free or overall survival. According to the authors the level is not relevant, please discuss the results from KM-plotter.

More discussion of the results from the different datasets is necessary. Results differ between the different datasets, the first show a result after 14 days while others do not show an increase the first 120 days. Did they all use tamoxifen or active metabolites? Was the last time point in these studies the final time point, if these were neoadjuvant studies the final time point would be the final surgery. Did the increase of ATF3 correlate with those who had progressive disease? If these datasets are from neoadjuvant treated patients are these then representative of the "normal" population that is treated with Tamoxifen. Neoadjuvant treated patients often have very large tumors and thus more heterogeneous tumors compared to Luminal A tumors.

The different datasets only have few patients included, it seems that the expression of ATF3 is quite the same in all patients, are these findings the same in the TCGA dataset? Are ATF3 expression in the same range in all datasets?

The resolution of the figures should be improved.

Reviewer 2 Report

Comments and Suggestions for Authors (will be shown to authors):

All in all this is an interesting article providing new data on the dynamic changes taking place in the early phases of tamoxifen resistance development with regards to ATF3. Also, the authors have done appropriate control experiments to show how down and upregulation of ATF3 influences sensitivity to tamoxifen and tumor growth in general. Clearly, endocrine resistance is a major clinical problem, where many different targeted drugs have been introduced in recent years to counteract this, but where we still need more knowledge and new compounds. As such, this paper will be of interest to oncologists and breast cancer researchers.

There are some issues that the authors need to consider, which are listed below. Also, they need to be more accurate when discussing the recurrence risk in hormone receptor positive breast cancer, this varies to a large degree depending on Tumor and Nodal stage as shown in the recent EBCTCG meta-analysis.

Title: The title should read "Time resolved profiling reveals ATF3 as a novel 2 mediator of tamoxifen resistance in breast cancer" since in almost all cases only TAM resistance was assessed, and they found no influence of letrozole on tumor group in mice when this was tested.

Abstract: This statement needs to be revised: "…Estrogen Receptor α (ERα) account for around 70% of cases and are mostly treated with targeted endocrine therapy. However, 40% of these tumors eventually relapse due to resistance development and further treatment of these patients is highly ineffective." This is simply not sure. All over, the recurrence-free survival after surgery for hormone receptor positive breast cancer is much higher than 60%. A 40% recurrence frequency would have to be for high-risk cases, typically with N+ disease. Also, the statement that further treatment of these patients when they recur is highly ineffective is not true. These patients in median have an overall survival of close to 4 years in the metastatic setting, with the use of endocrine therapy, CDK4/6 inhibitors ++

Introduction:

First paragraph: ER, PGR, HER2 and Ki67 are not factors used to define the luminal A, B subgroups, these are defined by gene expression analysis, such as PAM50. Please revise.

Line 65-66: resistance is critical,- …%:  in which patient group? early disease, advanced, metastatic?

Line 73-75 and 80-83,- the introduction introduces findings from the study, this part should be saved for the results.

Design:

Methods:

Sometimes the authors provide results with one cell line, sometimes with both  - were both cell lines used for all experiments? If not, they should present a table showing what experiments that was only performed with one cell line and the reason for choosing this one.

For instance: Transwell testing for Knockout: done on MCF7 (line 164-170), for overexpression they tested both MCF7 and T47D (line 216-221).

Nothing in the methods about the last result section (starts with line 289), regarding ATF3 increase upon endocrine adm. in patients. In particular,- no reference to GEO datasets,-should include a link and a methods section in how the different datasets where chosen. They should include a table of these studies with patients’ characteristics, type of  endocrine therapy given,- all in primary setting? or metastatic? TNM? And what type of gene expression analysis that was used.

Results:

All the figures in the article are blurry, sometimes the text is unreadable, the picture for transwell invasion assay in fig. 2 and 3 is a good example, so is the text in fig 1 b.

Abbreviations needs to be explained in the main text, not just suppl. info: for example, when using T47D-T and T47D-L (Fig. 1) this is only explained in the suppl. not in the main text. Same for OE, used one time, not explained,- line 206, all other places the write it out,- overexpressing.

It is stated that T47D and MCF7 are luminal A cell lines. No reference is given to work showing this for T47D, whereas for MCF7 reference 14 does not demonstrate this. Please provide references.

Figure 1e: WB show only one sample per cell line and condition, please provide at least 3 independent protein sample analyses of ATF3 and actin, and densitometry to show any significant differences.

Figure 2: Please specify whether the MCF7 cells used here are TAM resistant and whether they have been E2 deprived before starting the experiment. Also in the text, please explain why Crisp-Cas9 for ATF3 was not employed for T47D. Why was MCF7 chosen, and not T47D?

Figure 4: 4b: Please specify whether mice were given TAM + E2 versus E2 alone, or were they given TAM vs. E2, which would be the wrong comparison. 4c: Please indicate the number of mice per group, this should be at least 10 mice.

There is insufficient data provided to evaluate the response to letrozole in mice. First, they show no data demonstrating that the mice were postmenopausal or that their estrogen levels were sufficiently suppressed upon treatment to evaluate any potential benefit. Also, letrozole was tested in a very limited number of mice. The results with letrozole should therefore be removed.

Figure 5: The clinical findings are difficult to interpret without any data and separation of data for TAM vs. aromatase inhibitors, and pre- vs. postmenopausal women. Please provide. Also, more details regarding the six clinical materials are needed - what was different with the material where no change in ATF3 was found?

Based on the clinical materials that they have examined, they should provide data showing if an increase in ATF during endocrine therapy correlates with inferior survival or not.

Discussion:

The first sentence: "Resistance to endocrine therapy in breast cancer is an urgent clinical problem that affects around 40% of luminal breast cancer patients." This a very inaccurate statement - please specify whether this is patients with breast cancer recurrence after adjuvant therapy, or patients developing resistance in the metastatic setting, and please provide a reference showing this frequency of 40%.

The second sentence: "The limited understanding of the mechanisms underlying resistance 441 development requires greater efforts to identify new targetable genes and pathways involved in the process." This is also very inaccurate. There is actually a huge amount of research published on mechanisms of endocrine resistance. Please revise.

The authors state that letrozole was tested both in in vitro and in vivo. However, this referee can not find any data on testing in vitro. Please revise. And again, letrozole data are very limited in this paper (only one mouse experiment with findings that are impossible to interpret) - so letrozole data should be removed.

Round 2

Reviewer 2 Report

This reviewer´s comments have been addressed appropriately, and I would like to congratulate the authors on their important and nicely executed scientic work.